# Accumulated Disadvantage over the Lower Secondary School Years in Helsinki Metropolitan Area, Finland

**DOI:** 10.3390/ijerph17072290

**Published:** 2020-03-29

**Authors:** Sakari Karvonen, Laura Kestilä, Arja Rimpelä

**Affiliations:** 1Finnish Institute for Health and Welfare (THL), FI-00271 Helsinki, Finland; laura.kestila@thl.fi; 2Faculty of Social Sciences, Unit of Health Sciences, Tampere University, FI-33014 Tampere, Finland; arja.rimpela@tuni.fi

**Keywords:** multidimensional adversity, follow-up, social determinants, health, young people, comprehensive school, Finland

## Abstract

Accumulated disadvantage (AD) is conceptualised here as an agglomeration of unfavourable or prejudicial conditions which in adolescence may compromise the progress to further education or future life chances. There are several theories on AD, suggesting, e.g., (1) an increase of AD by age and (2) trajectories (previous disadvantage predicts later disadvantage). Social pathways theory suggests that (3) a third factor (e.g., socioeconomic position, SEP) mediates or moderates the association between early and later disadvantage, while other theories imply (4) polarisation (a strengthening association between AD and SEP by age) or (5) equalisation (a weakening of association between AD and SEP). We apply these theories to longitudinal data of 7th graders (13 years, *N* = 5742), followed until the end of the 9th grade. Five dimensions of disadvantage were health (poor self-rated health), social behaviour (poor prosocial behaviour), normative (conduct disorders), educational (poor academic achievement), and economic (parental unemployment). The results show that the prevalence of AD increased over the follow-up as most indicators of disadvantage elevated. AD at the 7th grade predicted later AD, as did the SEP of the students. Moderation of AD by SEP was also observed. The study corroborates with hypotheses on increase of AD, trajectory, and social pathways but no signs of polarisation or equalisation were observed.

## 1. Introduction

Accumulated disadvantage (AD) may be conceptualised as a simultaneous agglomeration of unfavourable or prejudicial conditions. In adolescence AD may compromise the progress to further education or the future life chances as it implies a lack of resources preventing access to a way of life considered normal in society, e.g., [1,2,3]. Earlier studies on young people, however, disregard the AD by focusing either on individual forms of disadvantage or certain groups considered to be risky or extreme, such as school drop-outs, the so-called NEETs (young people not in education, employment, or training) or youth from poverty-high neighbourhoods [4]. Relatively few studies have analysed AD empirically (exceptions include, however, [5,6]) and even fewer emphasise early adolescence even though adverse childhood conditions have been shown to be associated with several unfavourable outcomes later in life [7,8,9].

Given the relative lack of research on early adolescence, our interest here lies in assessing AD over the course of (comprehensive) lower secondary school, i.e., among those aged from 13 to 16, on average. The few earlier studies can be summed up to suggest that (1) accumulated disadvantage is heavily socially conditioned, (2) single forms of disadvantage tend to cluster (see [10,11]) and (3) school as a social environment may reinforce disadvantage [12]. However, most of these studies summarise findings based on single forms of disadvantage or have not covered adolescents still within the comprehensive school system. Yet this age is a particularly interesting phase of life given that many of the social roles are re-negotiated, and young people assume more independent and adult roles. This phase of life also stresses the importance of peer groups at school and during leisure time [13,14]. 

Following the Nordic welfare research tradition [1,15], we adopt here a multidimensional approach to disadvantage and analyse several forms of potential disadvantage, including health, behavioural problems in the social dimension (i.e., deficiencies in prosocial actions), normative (i.e., conduct disorders breaking social norms), and educational, as well as economic disadvantage. Our earlier study showed that when it comes to well-being and academic achievement, divergence between schools tends to be persistent [16], but whether this applies on the individual level remains uncertain. 

Unlike many other countries, in Finland a key transition in the educational system takes place after the 9th grade when students are selected either to an academic or a vocational track. In 2017, the majority (53%) continued their studies in the academic track and 41% followed the vocational one. The remaining share (6%) was split between those staying outside any education, those continuing in preparatory education of some form, and those not in the registry [17].

Given the multidimensionality of disadvantage, it is no surprise that several theories on AD are present in the literature. On the basis of these, we have identified the following five hypotheses. (1) increasing (accumulation of disadvantage over the school years), (2) trajectory (previous disadvantage predicting later disadvantage), (3) social pathway (moderation of the association between early and later disadvantage), (4) polarisation (strengthening association between AD and socioeconomic position over the school years), and (5) equalisation (weakening association between AD and socioeconomic position over the school years). The trajectory and social pathway hypotheses are derived from a more general life-course model. Furthermore, the theories exemplify somewhat different frameworks, thus, they are not completely mutually exclusive. For example, polarisation could be interpreted as a specific form of a social pathway: one leading to diverging pathways (according socioeconomic position). We tested the hypotheses in early adolescence using longitudinal data of 7th graders (13 years, *N* = 5742) from the Helsinki Metropolitan Area, followed up until the end of 9th grade (16 years) and by measuring five dimensions of disadvantage: health (poor self-rated health), social behaviour (prosocial behaviour disorders), educational (poor academic achievement), normative (conduct disorders), and economic (parent’s unemployment). (Table 1).

The hypothesis on an increase of AD is based on empirical findings showing that over the follow-up, single forms of disadvantage become more frequent. This is evident, e.g., in the cases of poverty [18] and poor health over age [22]. Thus, regarding our study, the hypothesis simply predicts that AD also becomes more frequent over age. Cumulative effects of different indicators are a specific focus of the life-course perspective out of which several possible hypotheses could be derived. Here, the trajectory model will be explored as it explicitly addresses accumulation of one risk factor, such as AD over time. Diane Kuh et al. [19] characterise a trajectory as “a long term view of one dimension of an individual’s life over time”, thus suggesting that an earlier form of disadvantage predicts later disadvantage of a similar form. While originally trajectory refers to specific forms of disadvantage and their predictive effects, here our focus is on accumulation. Hence, we have applied the concept of trajectory to assume that AD at the 7th grade is associated with AD at the 9th grade.

As part of the chain-of-risk models in the life-course perspective, it is acknowledged that earlier life effects may be mediated or moderated by external factors [19]. Following this idea, a third factor affects the relationship between earlier and later disadvantage, in other words, their trajectory. In our study, we refer to this hypothesis as a social pathway moderation hypothesis. According to this hypothesis, a third factor may influence the strength or the direction of the relationship between an independent (earlier AD) and a dependent variable (later AD), and operate as a moderator. This is the case, for example, when cumulative processes take a different pace or their paths vary in accordance with some socioeconomic variable or by gender. Technically, this suggests an interaction between independent variables in predicting the AD trajectory. The independent factors analysed here are socioeconomic variables (parent’s level of education, immigrant background, family type, area of residence) and gender.

The polarisation hypothesis, on the other hand, derives originally and famously from Robert Merton’s postulation of the so-called Matthew-effect [20]. Initially referring to the unequal distribution of rewards in science, the hypothesis has since been broadened to also include other processes generating inequality in society, in a way that means that the more advantaged groups receive more and those less well-off receive less (resources, power, status etc.). In our study, by following this idea, we assume that differences between the advantaged and the disadvantaged groups of young people enlarge over time, in other words, that the AD becomes more socially determined when students come of age.

The equalisation hypothesis represents an opposite assumption by stating that adolescence signifies a phase of life that is characterised by relative equality, meaning diminishing differences between socioeconomic groups. Originally this was postulated by Patrick West [21] to explain smaller health inequalities at a younger age. He proposed that (mandatory) school is an institution that effectively mixes young people from various social backgrounds thus implying a contextual (school-level) process, the outcome of which is a levelling off of diversities by social background. As adolescents pursue independence from their parents, they also renegotiate social hierarchies as a part of this social process. Here, equalisation hypothesis proposes that over time, social determinants become less important predictors of AD.

The main aim of the study is to analyse whether disadvantage accumulates over the course of (comprehensive) lower secondary school (age 13 to 16) and more specifically, to test if the hypotheses outlined above apply over this period of life. Finally, we expected gender to modify the associations [23,24].

## 2. Materials and Methods

The present study was based on three school surveys conducted in 2011 and 2014. Participants were 13-years-old at the baseline (7th grade) and they were recruited across comprehensive schools in the Helsinki Metropolitan Area in Southern Finland in 2011. The baseline survey was conducted in 2011 (in the 7th grade) and the follow-up in 2014 in the end of the 9th grade (ages 15–16 years), which is the final grade of the lower secondary school. The Helsinki Metropolitan Area covers 14 municipalities. A total of 7807 students who had participated in the first wave completed the follow-up survey (response rate 73%). The final data comprised 124 lower secondary schools from 14 municipalities (*N* = 5742). 

Our measure of disadvantage covered five dimensions: health, social behaviour, normative, educational, and economic. All measures were self-reported. The questions were identical in the questionnaires at the 7th and the 9th grade. The dependent variable, named accumulated disadvantage, was constructed by summing up the number of disadvantages and for the multivariate models the construct was dichotomised into those with two or more disadvantages (Table 2) and those with one or none. Already, Rutter [25] has shown that one form of disadvantage does not necessarily imply serious adversities while simultaneous existence of more forms may produce cascading adverse effects.

Health was measured with a single question covering self-rated health which is well-known to predict, e.g., mortality (e.g., [26]): “Do you think your health is…” with five response alternatives ranging from “very good” to “very poor”. To measure health disadvantage, the categories “very poor”, “poor” and “average” were combined. Two indicators were based on subscales of the Strengths and Difficulties Questionnaire (SDQ) [27]. The SDQ covers conduct disorders, emotional symptoms, hyperactivity and inattention, peer relationship problems, and prosocial behaviours. Its reliability and validity has been shown to be high [28]. For each item the options included a three point scale (0–2): “not true”, “somewhat true” or “certainly true”. Item ratings were added together to obtain a total score with lower scores indicating fewer problems. The subscales selected were prosocial behaviour and conduct problems in which an “abnormal” level was considered a threshold for (social or normative) disadvantage. Both scales ranged from 0 to 10. As the two scales were in reverse order, for prosocial behaviour the cut-off point was below 5 and for conduct problems the cut-off point was 5 or more problems. The validity of the subscales has been supported by earlier exploratory studies from several contexts (e.g., [28,29]) and they have been shown to be associated with relevant DSM-IV (Diagnostic and Statistical Manual of Mental Disorders) diagnoses even though self-reported scales are somewhat less well associated with diagnosed psychiatric disorders than those assessed by a teacher or parent [28].

Measurement of educational disadvantage is based on the Finnish grading system used for assessing students’ learning; the range is from 4 (fail) to 10 (excellent). The grade point average (GPA) was analysed by averaging responses to the question: “What was your school grade in your last report card?” Particular subjects such as math were inquired. Structured responses ranged from below 6.5 (lowest score) to 10 (highest score), with half-point categories. Thus the measurement echoed the Finnish grade system. In each category, the middle point was used as the GPA and those with less than 7.0 were included in the disadvantaged category. A fifth dimension of disadvantage was an economic one and it was based on parents’ employment status. The data did not include direct measures on students’ own economic status but given that at this age young people are still economically dependent on their families, we considered unemployment of the parents, reported by the child, to be the most accurate available data in reflecting the financial status of the children. The question was “Are your parents working?” with seven response options for both the mother and the father: “Working outside home”, “Working at home”, “At home, not working”, “Unemployed”, “Retired”, “Studying”, and “No father/mother”. With one or both parents unemployed, the students were considered to have an economic disadvantage.

Variation in the socio-economic background of the students was measured with the level of education of the parents, immigrant background, family structure, and area of residence. Educational level of parents refers to the highest education of the parents and was divided into three categories: lower, middle, and higher educational levels. Missing cases were excluded. Educational level was inquired by asking “What kind of education do your parents have?” with four response alternatives: “comprehensive school” (representing the lower level), “vocational degree” (middle level), and “matriculation” or “university or academic degree” (higher level). Immigrant background was dichotomised into those children with and without an immigrant background, determined by inquiring the language students normally use with their family or friends. Family structure was also dichotomised into intact families (those living with both parents) and other family types on the basis of the kind of family the students lived with most of the time. Area of residence was divided into three categories on the basis of the municipality of residence: capital area (four municipalities), surrounding regions of rapid growth (five municipalities), and other regions of the Metropolitan area (five municipalities) [30]. This categorisation focuses on regional division of labour—essentially reflecting the difference between the core and the periphery—which is especially relevant in a Metropolitan area. In such an area, the spatial location in relation to the core area also largely determines much of the social characteristics of the residents including their socioeconomic factors.

The main method of analysis was logistic regression analysis. Additionally, the data were analysed by cross-tabulations where significance of the differences was tested by Pearson’s chi-squared test (*p* < 0.05) and by using 95% confidence intervals. All individual-level analyses were stratified by gender as earlier studies suggested that the levels and to some extent also the pathways leading to disadvantage may differ between girls and boys [23]. The socioeconomic determinants of AD were studied to find out whether signs of equalisation or polarisation could be identified. Logistic regression models were performed separately for the baseline (7th grade) and the end point of the follow-up (9th grade). All analyses were performed with SPSS Statistics version 26.

## 3. Results

### 3.1. Descriptive Results

At the 7th grade, 63% of boys and 73% of girls did not report any form of disadvantage (Table 2). Among boys, the social behaviour dimension was most frequent with 17% reporting prosocial behaviour below the abnormal threshold. Among girls, the respective rate was only 5%. The highest frequency among girls (11%) was observed in poor health, however, no gender difference was observed. Equally, the gender difference was non-existent in the economic dimension while the other dimensions showed significant gender differences at 7th grade. In general, the levels were lowest at the educational dimension with 6% of boys and 5% of girls reporting poor grades. 

At the 9th grade, the highest rate was among boys in the educational dimension with almost a fifth reporting poor educational performance. Poor health was also frequent with 15% of both genders reporting poor self-rated health. Lowest rates were found among girls in the social behaviour dimension (5% showing signs of deviant behaviour) and in the normative dimension (8%).

Statistically significant gender differences were found at the social behaviour, normative, and educational dimensions both at the 7th grade and at the 9th grade (Table 2).

### 3.2. The Increase Hypothesis

To test the increase hypothesis, a score summing up two or more forms of disadvantage (accumulated disadvantage) was constructed. At the 7th grade, 9% of boys and 5% of girls had accumulated disadvantage and over the follow-up these levels doubled showing an increase of AD (Table 2). Of the single indicators of disadvantage, apart from indicators of the social behaviour dimension and the economic one for girls, all had an elevated level at the 9th grade compared to the 7th grade.

### 3.3. The Trajectory Hypothesis

The trajectory hypothesis was first tested by analysing the pairwise association between AD at the 7th grade and at the 9th grade. Table 3 also shows the pairwise associations between individual indicators of disadvantage in the follow-up. All associations were statistically significant, and the strongest ones were those of AD’s (OR = 6.19) and of poor health (OR = 6.39). 

Next, AD at the 7th grade was included in a model accounting for AD at the 9th grade. This was performed to assess the extent to which previous disadvantage determines disadvantage observed at a later stage as suggested by the trajectory hypothesis Table 5. Unsurprisingly, earlier disadvantage was the strongest (OR = 4.76, *p* < 0.001) predictor of AD at the end of the follow-up supporting the trajectory hypothesis further. However, controlling for earlier disadvantage only had a minor effect on the associations with the other determinants. Thus, all socioeconomic measures along with gender retained a statistically significant association with AD.

### 3.4. The Social Pathway (Moderation) Hypothesis

The significance of interactions between socioeconomic background and AD at the 7th grade in predicting AD at the 9th grade was tested to assess the social pathway hypothesis. A significant interaction was interpreted to support the hypothesis. The analysis showed two interactions: between gender and early AD as well as between parents’ education and early AD. These are illustrated in Appendix A showing frequencies in the (cross-tabulated) categories of the three variables. The table shows that for those with no accumulated disadvantage at the 7th grade, AD at the 9th grade was highest for those whose parents had the lowest level of education whereas for those with AD already at the 7th grade, the association was reversed for girls. Among boys with AD at the 7th grade, there was no association with parents’ education.

### 3.5. The Equalisation and Polarisation Hypotheses

All variables in the fully-adjusted models had a statistically significant association with AD both at the 7th grade and at the 9th grade (Table 4). Students from families with the lowest level of education, with immigrant backgrounds and with non-intact families were more likely to report AD. Furthermore, students residing either in areas of rapid growth or other regions than the capital area had higher levels of AD. As the associations of socioeconomic factors with AD remained at the same level in both grades (Table 3), no support for either equalisation or polarisation hypotheses was observed.

Secondly, socioeconomic determinants were studied for the end point by controlling the level of accumulated disadvantage at the 7th grade (Table 5). In addition to socioeconomic measures, gender was included into the models to account for the different levels of disadvantage between girls and boys.

Overall, given that the socioeconomic effects of AD remained the same over time, there was little support for the equalisation hypothesis. Yet, the polarisation hypothesis was not supported either as suggested by the same evidence. In other words, despite a higher level of AD at the end of the follow-up, AD did not become socially more distinctive.

## 4. Discussion

The emphasis of this study was to gain a better understanding of accumulation processes that take place in early adolescence. This age group is still within the comprehensive school system. From the point of view of accumulation of disadvantage, this group is under researched. We reviewed some key theories of AD and came up with five hypotheses on possible mechanisms of accumulation which are labelled as follows: increase by age, trajectory, social pathway (moderation), polarisation and equalisation. Our empirical data on students during the school years from the 7th grade to the 9th grade provided support for the increase, trajectory, and to some extent for the moderation of social pathway hypotheses. No support for polarisation or equalisation hypotheses was found.

From the theoretical point of view, the contribution of the study is two-fold. On one hand, it contributes to the literature aiming to specify and explain how different mechanisms underlying disadvantage develop. As Schafer et al. [31] point out, more empirical research is needed on processes through which early disadvantage leads to different outcomes. Our study shows that an increase of AD and a trajectory of AD are typical for early adolescence. In fact, a large proportion of those with AD at the 9th grade were already subject to it at the 7th grade. Additionally, pathways to AD are modified by gender and parents’ level of education. This shows that structural elements operate early over the course of an adolescents’ life and over a relatively short period of life. Furthermore, according to this study, the comprehensive school is not able to level these off, even if the original differences did not polarise further.

On the other hand, the study contributes to the theories that stress dimensionality of disadvantage. The maxim that disadvantage accumulates has guided much of empirical research [31]. We show here that early adolescence is a critical period of life for the onset of many dimensions of disadvantage. Consequently, towards the end of early adolescence, a larger number of young people are exposed to multiple risk factors and thus have a greater risk for poor adjustment to life adversities than exposure to single disadvantages. In other words, the risks are multiplied as a consequence of them cascading [25]. We found the prevalence of accumulated disadvantage to increase clearly with age thus verifying the increase hypothesis. The prevalence of most single forms of disadvantage increased during the follow-up as well. The increase was also evident with the rate of AD which doubled among boys (9% vs. 18%) as well as among girls (5% vs. 10%). Almost a fifth of the boys had at least two forms of disadvantage at the 9th grade. An additional analysis showed that a fourth of the students (25%) comprised of those whose disadvantage had accumulated already at the 7th grade representing 46% of the initially disadvantaged group.

Of the single sociodemographic indicators, gender differences were widest in social behaviour and in the normative dimension. Poor health was relatively common at this age, too, but gender difference was not observed. Potentially, there are several processes accounting for these changes. Some of the patterns found may result from pubertal changes occurring “naturally”, or from their social correlates. This explanation has been suggested to reveal why among males conduct-related behavioural dimensions start to predominate while among females internalised disorders become more frequent. An alternative interpretation refers to the comprehensive school as a social environment and points to the gendered patterns of the formal and the informal school [32,33]. Research supporting this notion has shown that certain behaviours may be socially tolerated for one gender but less for the other. 

Equalisation and polarisation hypotheses predicts opposite patterns. The analyses showed that AD was socially heavily distinctive already at the 7th grade, but the impact of socio-economic background remained the same over time. Support was not found for equalisation nor polarisation. Analyses of interactions between early AD and the socio-economic factors (gender, level of education of parents), however, showed that the effects of earlier disadvantage are moderated by parents’ education, which corroborates with the social pathway hypothesis. Other studies have shown that parental academic involvement in middle school may alleviate problem behaviours and generate higher educational aspirations and that these effects may depend on SEP [34].

The trajectory hypothesis was supported, too. We found that the initial disadvantage predicted later AD strongest, even after controlling for the socio-economic factors. Earlier studies have shown that various forms of disadvantage experienced in childhood generate disadvantages later in adulthood (e.g., [7]), but here we have revealed that already over the course of early adolescence many of the key mechanisms by which disadvantage accumulates are in effect. Our study contributes to this research by identifying the paths that operate while students are still within the comprehensive school. Levelling off the differences in opportunity structures is among the overt aims of this phase of schooling. 

We focused on early adolescence, which is a relatively neglected phase of life in studies of AD. The follow-up covered three school years at the end of comprehensive education, which in the Finnish school system comprises grades 7 to 9, ages 13 to 16 years. These years are in many ways decisive for a student’s future educational trajectories. The school marks (grade point average) determine to a large degree whether the adolescent is able to select an academic path, a vocational path, is left without a place in upper secondary school, or has to delay in starting studies. In countries where the academic divergence takes place at an earlier age, the effects may be even stronger.

Future studies should analyse whether the patterns observed within the comprehensive school system can be observed in societies with other educational contexts, especially those where the educational paths diverge earlier. Furthermore, the level of AD may differ, e.g., depending on the welfare system.

To our knowledge, this is the first study that follows accumulation of disadvantage at early adolescence and tests hypotheses on accumulation this comprehensively. The strengths of the study include a large data set that covers practically the whole population of the area thus resembling a total sample. Furthermore, the prospective follow-up data include a unique set of themes that allowed us to combine variables related to learning, well-being and social indicators. A limitation is that only about two thirds of those who answered the baseline survey could be traced in the follow-up. The participation was also voluntary for schools, which is why all of them did not conduct the survey. Some students were absent, changed school, or did not fill in the questionnaire. This may distort results but usually associations between variables do not change even though their strength may diminish or strengthen.

A further limitation is the narrow age span covering three school years in the lower secondary school. On the other hand, in terms of life-course, this phase is pivotal because it is crucial for the selection of the following education career as well as for adolescent physiological, mental, and social development. We were not able to measure the economic status of young people directly. Instead, we adopted a measure based on the employment status of the parents. Other studies have shown that in Finland the family economic status—clearly influenced by unemployment in the family—is reflected in an adolescents’ own economic status [35]. At this age, young people are rarely employed themselves nor do they possess other economic means that would allow them economic independence from their families. Finally, the data derives from a predominantly urban population which means that the findings do not necessarily reflect the whole adolescent population. Furthermore, analysing school context was beyond the boundaries of this study. Other studies have shown that class level especially is an important factor differentiating between students. Our data did not, however, cover class level data.

## 5. Conclusions

The study shows that disadvantage accumulates already in early adolescence and that the prevalence of experiencing it increases by age. Even though accumulated disadvantage varied between sociodemographic groups, no polarisation or equalisation in disadvantage between the groups was noticed during the lower secondary school years. On one hand, this means that school as an institution has not been able diminish the differences but, on the other, it has not contributed to their increase either. The results may also reflect the fact that the school system is not able to compensate the differences in well-being created by the family of origin and earlier school years. The study suggests that already at an early age accumulated disadvantage increases over time and is based on trajectories and risks deriving from earlier life.

## Figures and Tables

**Table 1 ijerph-17-02290-t001:** Key theories of accumulated disadvantage (AD) and hypotheses derived from them and applied to school years between 13 to 16 years.

Theory	Hypothesis
Increase of AD [18]: More cumulative disadvantage over time	Accumulated disadvantage becomes more frequent over the school years
Trajectory [19]: One form of earlier disadvantage predicts later disadvantage of a similar form	Previous accumulated disadvantage associates with later accumulated disadvantage
Social pathway: moderation [19]: The association between earlier and later accumulated disadvantage is moderated by independent factors leading to diverging trajectories	The effect of earlier AD on later AD is moderated by a socio-economic factors (parents’ level of education, family type, area of residence, immigrant background) or gender
Polarisation [20]: ‘Matthew-effect’: Increasing differences between privileged vs less privileged groups	Stronger association between socioeconomic position (SEP) and accumulated disadvantage over time
Equalisation [21]: Social differentiation between socioeconomic groups level off over the school years	Weakening association between accumulated disadvantage and SEP during school years

**Table 2 ijerph-17-02290-t002:** Prevalence (%) of dimensions of disadvantage and accumulated disadvantage at the 7th and 9th grades, by gender (*N* = 4337–5677) and their statistical significance (*p*) by gender and grade (95% confidence interval).

Dimension and Indicator of Disadvantage	Boys		Girls		*p* between Genders	
	7th	9th	7th	9th	7th	9th
Health: Poor or average self-rated health	**11.7 ^1^**	**15.4**	**10.8**	**15.2**	0.295	0.828
Social Behaviour: SDQ ^2^: abnormal prosocial behaviour	16.9	16.4	5.2	4.9	<0.001	<0.001
Normative: SDQ ^2^: abnormal conduct	**7.3**	**15.6**	**5.1**	**7.9**	0.001	<0.001
Economic: Parents’ unemployment	**7.6**	**10.8**	8.6	10.0	0.172	0.346
Educational: GPA ^3^ below 7	**6.3**	**17.6**	**4.7**	**11.2**	0.012	<0.001
Not Accumulated: One or none of the indicators	**89.3**	**79.2**	**94.0**	**89.2**	<0.001	<0.001
Accumulated: Two or more indicators	**10.8**	**20.8**	**6.3**	**10.8**	<0.001	<0.001

^1^ Bold = 95% confidence intervals do not overlap; ^2^ Strengths and Difficulties Questionnaire; ^3^ Grade Point Average.

**Table 3 ijerph-17-02290-t003:** Odds ratios (OR, 95% CI: Confidence Interval) for each indicator of disadvantage at the 9th grade when having the same disadvantage at the 7th grade. (*N* = 4239–5677).

Indicators of Disadvantage at 7th Grade	9th Grade, OR	*p*
Health: Poor or average self-rated health	6.39 (5.34–7.64)	<0.001
Social Behaviour: SDQ ^1^: abnormal prosocial behaviour	1.63 (1.29–2.06)	<0.001
Normative: SDQ ^1^: abnormal conduct	1.33 (1.05–1.69)	0.017
Economic: One of the parents unemployed or stays at home	1.47 (1.14–1.89)	0.003
Educational: GPA ^2^ below 7	1.68 (1.36–2.07)	<0.001
Has accumulated disadvantage at 7th grade	6.19 (4.91–7.81)	< 0.001

^1^ Strengths and Difficulties Questionnaire; ^2^ Grade Point Average.

**Table 4 ijerph-17-02290-t004:** Sociodemographic factors accounting for accumulated disadvantage at the 7th and at the 9th grades. Odds ratios (OR, 95% CI: Confidence Interval) and *p*-values, bivariate models.

Sociodemographic Factor	7th Grade, OR	*p*	9th Grade, OR	*p*
Gender (girls)				
- boys	1.84 (1.47–2.30)	<0.001	2.19 (1.86–2.57)	<0.001
Parents’ level of education (high)				
- middle	1.27 (0.97–1.65)		1.29 (1.07–1.55)	
- lower	2.17 (1.58–2.97)	<0.001	2.20 (1.74–2.77)	<0.001
Immigrant background (no)				
- yes	1.73 (1.18–2.56)	0.006	1.73 (1.30–2.30)	<0.001
Family type (intact)				
- other	2.02 (1.62–2.52)	<0.001	1.80 (1.53–2.11)	<0.001
Area of residence (capital area)				
- area of rapid growth	1.30 (1.00–1.67)		1.50 (1.24–1.81)	
- other areas	1.63 (1.22–2.17)	0.002	1.86 (1.51–2.29)	<0.001

**Table 5 ijerph-17-02290-t005:** Odds ratios (OR) of sociodemographic factors accounting for accumulated disadvantage at the 9th grade. Logistic regression analysis. (*N* = 4183).

Sociodemographic Factor	9th Grade, OR	*p*
Gender (girls)		
- boys	1.90 (1.58–2.28)	<0.001
Parents’ level of education (high)		
- middle	1.28 (1.03–1.59)	
- lower	2.03 (1.54–2.65)	<0.001
Immigrant background (no)		
- yes	1.53 (1.08–2.16)	0.016
Family type (intact)		
- other	1.56 (1.29–1.89)	<0.001
Area of residence (capital area)		
- area of rapid growth	1.42 (1.14–1.76)	
- other areas	1.96 (1.55–2.49)	<0.001
Accumulated disadvantage at grade 7 (no)		
- yes	4.76 (3.72–6.08)	<0.001

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
