# Peer review of "Accumulated Disadvantage over the Lower Secondary School Years in Helsinki Metropolitan Area, Finland"

_ijerph, 2020, doi:10.3390/ijerph17072290_

Round 1

Reviewer 1 Report

The authors present an investigation about the association between the theories on accumulated disadvantage (AD) and their application in secondary school children divided by gender.

The introduction section is rather long. The paragraph 3 (Lines 74-78) should be changed to another place, e.g., to line 55, before explaining the theories on AD.

Method:

- Data was collected 6 years ago. I suggest to have a small subsample that is more recent to see the current situation. Or another measure that shows the possibility of maintaining the variables(e.g., specifying in the title/aim that the data and results would correspond and be applicable to that specific population).

- Statistical analysis: mention the level used of significance, as well as the program used for the statistical analysis.

Results:

- Table 2: provide numerical p-values as done in the other tables.

- Tables: explain abbreviations in notes underneath the tables.

- Lines 244-246: the sentence should be stated in the method section instead.

The discussion section is also long and hard to read.

Be consistent in the explanation of abbreviations at first use in all the text of the manuscript.

Keywords: erase words used in title, and replace by new ones to increase visibility of the article.

Reviewer 2 Report

This manuscript has a focused research frame and it concerns understudied field of accumulated disadvantage. Both the literature review and the analysis present new and considerable tools and outcomes to the research done within this area. There are only few minor modifications I want to suggest.

1) The analysis is well performed and precise. Only factor that seems is undefined is ‘student resident’, which is classified as ‘rapid growth’, ‘other regions [than the capital area]’ and ‘capital area’. What is the hypothesis behind this classification? How it relates to the accumulation of disadvantage? Would there be other options for the residential area indicators, for instance the socio-economic statistics of the municipal? I believe that the socio-economic composition of the local neighborhood would tell something more about the cumulative nature of disadvantage. In all, the role of the municipality of residence is detached and inarticulated.

2) Other factor that could be developed is the composition of the schools /classes. There is great variety of selectivism at the school-level in the Metropolitan area, and the class-level mixing of students differs. Is it possible to add the school or class-level factor for the analysis? If not (or it does not fit into this article), the authors could elaborate this a bit in the Discussion: the comprehensive school is not that comprehensive in Finland as is assumed in the article. If in some municipalities almost 30 % of lower secondary pupils are in selective classes, the equalization or polarization effect might prevail in some classes / schools / municipalities although at the average level there was no signs of it.

3) The analysis is well conducted and clearly reported.

I believe there is a mistake in the 3.2. (Descriptive results), row 4. Should the highest frequency among girls be 11% (instead of 12%)?

There is also missing (at least in my prints), the label ‘ns.’ from the Table 2., ECONOMIC row, last column.

The OR 4.7. under the heading of 3.3. (The trajectory hypothesis), second paragraph, concerning earlier disadvantage, is missing one decimal.

4) My last notions concerns concepts and definitions.

In the introduction, the authors suggest that the ‘AD may compromise the progress to higher education or the future life changes’… I believe the ‘higher education’ here means further education or further studies, not higher education (as tertiary education)? Since the vocational education diploma is not an indicator of disadvantage, it might be more precise to use another term than higher education. How about further studies / further education (although this also means non-tertiary education)?

Officially there are no ‘academic’ and ‘vocational’ ‘tracks’ in Finland, but general upper secondary education and vocational education and training (VET). Since both give qualifications to tertiary education, it might be politically important to label them more precise (at least general instead of academic).

‘Prosocial behavior’ is labelled as ‘social disadvantage’. Although I understand the authors need short labels for the tables, one might expect ‘social disadvantage’ to refer to social resources, for instance. Would the ‘social behavior’ be too long label?

‘Ethnicity’ is labelled here as ‘immigrant background’, and measured with the language that the pupils use (non-Finnish?). These are not equal. I would at least suggest excluding the concept of ‘ethnicity’.

Round 2

Reviewer 1 Report

I believe that the authors made the changes that were requested. And that the article it is ready to be published.